# Optimization Framework for Semi-supervised Attributed Graph Coarsening

**Manoj Kumar**[*,1]    **Subhanu Halder**[*,1]    **Archit Kane**[2]    **Ruchir Gupta**[2]    **Sandeep Kumar**[1]

[1]Dept. of Electrical Engineering, Indian Institute of Technology Delhi, India
[2]Dept. of Computer Science and Engineering, Indian Institute of Technology (BHU) Varanasi, India

## Abstract

In data-intensive applications, graphs serve as foundational structures across various domains. However, the increasing size of datasets poses significant challenges to performing downstream tasks. To address this problem, techniques such as graph coarsening, condensation, and summarization have been developed to create a coarsened graph while preserving important properties of the original graph by considering both the graph matrix and the feature or attribute matrix of the original graph as inputs. However, existing graph coarsening techniques often neglect the label information during the coarsening process, which can result in a lower quality coarsened graph and limit its suitability for downstream tasks. To overcome this limitation, we introduce the Label-Aware Graph Coarsening (LAGC) algorithm, a semi-supervised approach that incorporates the graph matrix, feature matrix, and some of the node label information to learn a coarsened graph. Our proposed formulation is a non-convex optimization problem that is efficiently solved using block successive upper bound minimization(BSUM) technique, and it is provably convergent. Our extensive results demonstrate that the LAGC algorithm outperforms the existing state-of-the-art method by a significant margin.

## 1 INTRODUCTION

Graphs, as foundational mathematical structures, hold immense significance across diverse domains such as material science, finance, biology, and chemistry [Battaglia et al., 2018, Wu et al., 2020, Zhou et al., 2020, Bruna et al., 2013, Chen et al., 2020, Defferrard et al., 2016]. Serving both as end goals and preprocessing tasks for various models,

---

[*]These authors contributed equally to this work

graphs play a pivotal role in representing and analyzing intricate relationships within datasets [Kumar et al., 2020]. Nevertheless, the ever-increasing size of datasets presents a formidable challenge, requiring substantial memory resources and computational power to execute downstream tasks effectively [Chen et al., 2022]. This growing scale underscores the critical need for innovative approaches and optimizations to harness the full potential of graph-based analyses in today's data-intensive landscape.

In response to these challenges, the landscape has seen the emergence of techniques like graph coarsening[Loukas and Vandergheynst, 2018, Loukas, 2019, Kumar et al., 2023a,b,a, Dorfler and Bullo, 2011, Ron et al., 2010, Hendrickson et al., 1995], graph condensation [Jin et al., 2021], and graph summarization [Riondato et al., 2017]. These innovative approaches are designed to learn a smaller and more tractable graph while retaining the properties of the original graph.

There exist various graph reduction techniques. The most recent are: Loukas and Vandergheynst [2018], Loukas [2019] are heuristic-based approaches, Jin et al. [2021] is a deep learning-based technique, Kumar et al. [2023a] is an optimization-based framework. However, Loukas and Vandergheynst [2018], Loukas [2019] considers only the Laplacian of original graph while Jin et al. [2021], Kumar et al. [2023a] considers Laplacian matrix as well as feature matrix of the original graph for learning a coarsened graph.

These methods focus on learning a crucial mapping matrix to connect nodes in the original graph to supernodes in coarsened graphs. For a given original graph, multiple coarsened graphs can be generated. To assess the quality of a coarsened graph, the node profile matrix is introduced, as detailed in section 3.1 [Ghoroghchian et al., 2021]. This matrix, relying on the mapping matrix and the one-hot label matrix of the original graph, is essential for achieving a well-balanced mapping. To ensure an optimal coarsened graph for downstream tasks, the node profile matrix of the coarsened graph ideally should exhibit maximum sparsity. However, existing graph coarsening methods are not able

to learn coarsened graphs with sparse $\phi$ matrices, limiting their effectiveness for downstream tasks.

To enhance downstream task efficacy with coarsened graphs, achieving a sparse node profile matrix is crucial. In this paper, we propose an optimization-based method incorporating a function dependent on the mapping matrix C and a one hot matrix of some of the node labels of the original graph. The proposed formulation also includes Dirichlet energy and log determinant, constituting a non-convex optimization problem efficiently solvable through block successive upper bound minimization(BSUM) technique. We present the Label-Aware Graph Coarsening (LAGC) algorithm, updating variables iteratively, one at a time, while keeping others constant. Our algorithm is proven convergent, providing a robust and efficient solution to the optimization problem.

To demonstrate the efficacy of our algorithm, we applied it to a downstream task—specifically, node classification and link prediction using the coarsened graph. Utilizing the LAGC algorithm, we learned the coarsened graph, considering the graph matrix, feature matrix, and some of the node labels from the original graph. Subsequently, we trained a Graph Neural Network (GNN) using the learned coarsened graph. Testing was then conducted on the original graph. Notably, our results exhibited a substantial performance improvement over existing state-of-the-art methods, underscoring the superior capabilities of our proposed approach.

Our main contributions can be summarized as follows:

1. This is the first optimization method that leverages the graph matrix, feature matrix, and label matrix of the original graph to learn a more informative coarsened graph, optimizing its suitability for downstream tasks.

2. The proposed method is an efficiently solvable optimization technique utilizing block successive upper bound minimization(BSUM) technique, updating one variable at a time while maintaining the other fixed. Additionally, the method is proven to be convergent.

3. To demonstrate the effectiveness of our proposed algorithm, we conducted a downstream task, specifically node classification and link prediction. We trained a Graph Neural Network (GNN) using the coarsened graph, and testing was carried out on the original graphs. It is clear that our LAGC algorithm outperforms the state-of-the-art method significantly.

## 1.1 OUTLINE AND NOTATION

The paper is organized as follows: in Section 2, we present foundational background information covering graphs, graph learning from data, and graph coarsening techniques. Additionally, we introduce the proposed LAGC formulation in this section. Section 3 is dedicated to the development

of our algorithm. Finally, in Section 4, we present the outcomes of our experiments conducted on real-world datasets. In terms of notation, lower case (bold) letters denote scalars (vectors) and upper case letters denote matrices. The dimension of a matrix is omitted whenever it is clear from the context. The $(i, j)$-th entry of a matrix $X$ is denoted by $X_{ij}$. $X^{\dagger}$ and $X^{\top}$ denote the pseudo inverse and transpose of matrix $X$, respectively. $X_i$ and $[X^T]_j$ denote the $i$-th column and $j$-th row of matrix $X$. The all-zero and all-one vectors or matrices of appropriate sizes are denoted by $\mathbf{0}$ and $\mathbf{1}$, respectively. The $\|X\|_1, \|X\|_F, \|X\|_{1,2}$ denote the $\ell_1$-norm, Frobenius norm and $\ell_{1,2}$-norm of $X$, respectively. The Euclidean norm of the vector $X$ is denoted as $\|X\|_2$. $\det(X)$ is defined as the generalized determinant of a positive definite matrix $X$, i.e., the product of its non-zero eigenvalues. The inner product of two matrices is defined as $\langle X, Y \rangle = \text{tr}(X^{\top}Y)$, where $\text{tr}(\cdot)$ is the trace operator. $\mathbb{R}_+$ represents positive real numbers. The inner product of two vectors is defined as $\langle X_i, X_j \rangle = X_i^T X_j$ where $X_i$ and $X_j$ are the $i$-th and $j$-th column of matrix $X$.

## 2 BACKGROUND AND PROBLEM FORMULATION

In this section, we review the basics of graph and graph coarsening.

## 2.1 GRAPH

A graph with features and labels is represented as $\mathcal{G} = (V, E, A, X, Y)$, where $V = \{v^1, v^2, ..., v^p\}$ denotes the vertex set, $E \subseteq V \times V$ is the edge set, and $A \in \mathbb{R}_+^{|\times|}$ stands for the adjacency (weight) matrix for a graph having $p$ number of nodes. Each non zero entry $A_{ij}$ represents the edge between the $i^{th}$ and $j^{th}$ nodes. Furthermore $X \in \mathbb{R}^{p \times n} = [\mathbf{x}_1, \mathbf{x}_2, \ldots, \mathbf{x}_p]^{\top}$ is a feature matrix, where each row vector $\mathbf{x}_i \in \mathbb{R}^n$ represents the feature vector associated with one of the $p$ nodes of the graph $\mathcal{G}$. Moreover, in semisupervised learning, label information is given by $Y \in \{0, 1\}^{p \times l}$, where if node $v^i$ is labelled, then $\mathbf{y}_i :$ represents the corresponding one-hot indicator vector; otherwise, $\mathbf{y}_i := 0$ for unlabelled data. In the general, a graph is represented by either an adjacency matrix or a Laplacian matrix. A matrix $\Theta \in \mathbb{R}^{p \times p}$ is identified as a combinatorial Laplacian matrix when it belongs to the following set [Kumar et al., 2020]:

$$\mathcal{S}_{\Theta} = \Big\{ \Theta_{ij} = \Theta_{ji} \leq 0 \text{ for } i \neq j; \Theta_{ii} = -\sum_{j \neq i} \Theta_{ij} \Big\}. \quad (1)$$

Moving forward, the relationship between the adjacency matrix $A$ and the combinatorial Laplacian matrix is defined as $A_{ij} = -\Theta_{ij}$ for all $i \neq j$, and $A_{ij} = 0$ for $i = j$.

Highlighting the advantages of the Laplacian matrix $\Theta$ over the adjacency matrix $A$, $\Theta$ possesses key properties such as being a positive semidefinite matrix, a symmetric matrix, and having zero row sums. In the subsequent subsection, we will delve into a discussion on graph learning from data.

## 2.2 GRAPH LEARNING FROM DATA

Given the data $X = [\mathbf{x}_1, ..., \mathbf{x}_p]^T$, a connected and smooth graph can be obtained by solving the following optimization problem [Kalofolias, 2016]:

$$\min_{\Theta \in \mathcal{S}_\Theta} -\gamma \log(\det(\Theta + J)) + \text{tr}(X^T \Theta X) + \beta h(\Theta) \quad (2)$$

where, $\Theta \in \mathbb{R}^{p \times p}$ represents the target Laplacian matrix, and $\mathcal{S}_\Theta$ is the set of Laplacian matrices as defined in (1). The term $\text{tr}(X^T \Theta X)$ represents the smoothness or energy of the graph, and minimizing it signifies that nodes with similar features will have higher edge weights. Next, $\beta$ is a hyperparameter, and the regularizer $h(\Theta)$ enforces desired properties e.g. sparsity in the coarsened graph. Ensuring the connectedness of the graph requires maintaining the rank of $\Theta$ as p-1. This is achieved through the term $-\gamma \log(\det(\Theta + J))$, where $J = \frac{1}{p}\mathbf{1}_{p \times p}$ is a rank-1 matrix with each element equal to $\frac{1}{p}$. The addition of $J$ to $\Theta$ ensures a full-rank matrix without altering the row and column space of $\Theta$. Next, in the subsequent subsection, we will delve into the discussion of graph coarsening.

## 2.3 GRAPH COARSENING

The objective of graph coarsening is to learn a smaller, more tractable graph $\mathcal{G}_c(\Theta_c, \tilde{X}, \tilde{Y})$ while preserving the properties of the original graph $\mathcal{G}(\Theta, X, Y)$. Where, $\Theta_c \in \mathbb{R}^{k \times k}$ is the Laplacian matrix, $\tilde{X} \in \mathbb{R}^{k \times n}$ is the feature matrix, $\tilde{Y} \in \mathbb{R}^{k \times l}$ is the label matrix of the coarsened graph. The relation between, $\Theta$ and $\Theta_c$, $X$ and $\tilde{X}$, $Y$ and $\tilde{Y}$ are given by,

$$\Theta_c = C^T \Theta C, \quad X = C\tilde{X}, \quad \tilde{Y} = \text{argmax}(C^\dagger Y) \quad (3)$$

Where $C \in \mathbb{R}_+^{p \times k}$ is the mapping matrix that maps the $p$ number of nodes of original graph to $k$ number of nodes of the coarsened graph. Also, each non zero entry of $C$ i.e. $C_{ij}$ indicate $i^{th}$ node of original graph get mapped to the $j^{th}$ super node of the coarsened graph. For a balanced mapping, the mapping matrix must belong to the following set:

$$\mathcal{C} = \Big\{ C \geq 0 | \langle C_i, C_j \rangle = 0 \,\forall\, i \neq j, \quad \langle C_i, C_i \rangle = d_i,$$
$$\|C_i\|_0 \geq 1 \text{ and } \big\|[C^\top]_i\big\|_0 = 1 \Big\} \quad (4)$$

**Problem Statement:** Given an original graph $\mathcal{G}(\Theta, X, Y)$, our objective is to learn a coarsened graph $\mathcal{G}_c(\Theta_c, \tilde{X}, \tilde{Y})$.

Several graph coarsening techniques have been developed for learning the mapping matrix $C$. The heuristic method proposed by [Loukas, 2019] focuses solely on the Laplacian matrix $\Theta$ to derive the mapping matrix $C$. In contrast, [Jin et al., 2021] is a deep learning based method leveraging graph neural networks for learning a condensed graph. A more recent and comprehensive optimization-based method is introduced by [Kumar et al., 2023a]. This method not only considers the Laplacian matrix $\Theta$ but also incorporates the feature matrix $X$ for the learning of the mapping matrix $C$.

In semi-supervised learning, where some node label information is available, existing state-of-the-art methods often neglect the label information during the coarsening process or, equivalently, while learning the mapping matrix $C$. This oversight in utilizing the label matrix might result in the learning of a less informative coarsened graph, rendering it unsuitable for downstream tasks and potentially undermining the purpose of the coarsening process. In response, we introduce the first framework that incorporates the feature matrix $X$, label matrix $Y$ in which some node labels are known, and Laplacian matrix $\Theta$ of the original graph as inputs in the learning of a coarsened graph. The proposed formulation is:

$$\min_{\Theta_c, \tilde{X}, C} f(\Theta_c, \tilde{X}) + \beta h(\Theta_c) + \frac{\lambda}{2} g(C) + r(C, Y) \quad (5)$$

$$\text{s.t.} \quad C \geq 0, \ \Theta_c = C^T \Theta C, \ X = C\tilde{X}, \ \Theta_c \in \mathcal{S}_\Theta, C \in \mathcal{C}$$

where, $f(\Theta_c, \tilde{X})$ is a graph fitting term, for example $f(\Theta_c, \tilde{X}) = \text{tr}(\tilde{X}^T \Theta_c \tilde{X})$ represents the smoothness of the graph. Subsequently, the regularizer $h(\Theta_c)$ is applied to the Laplacian matrix to ensure crucial properties in the graph. For instance, $h(\Theta_c) = \|C^T \Theta C\|_F^2$ is employed to ensure sparsity in the resulting coarsened graph. This regularization term contributes to shaping the graph by imposing constraints that lead to a more structured and meaningful representation. Additionally, the regularizer $g(C)$ is imposed on the mapping matrix $C$ to enforce desired properties outlined in $C$ as defined in (4).

Subsequently, the function $r(C, Y)$ plays a pivotal role in our approach, incorporating the label matrix $Y$ of the original graph and the mapping matrix $C$. This function maps nodes with similar labels in the original graph to a supernode in the coarsened graph. The careful selection of the function $r(C, Y)$ is paramount as it directly influences the quality of the coarsened graph. Determining the appropriate function is a challenging task. In the next subsection, we will delve into the algorithm development.

## 3 ALGORITHM DEVELOPMENT

Before delving into the algorithm development, we will explore the concept of the node profile matrix [Ghoroghchian et al., 2021] in the subsequent section.

## 3.1 NODE PROFILE MATRIX

Given an original graph $\mathcal{G}(\Theta, X, Y)$, the objective of graph coarsening is to learn a coarsened graph $\mathcal{G}_c$ through the learning of the mapping matrix $C$. The quality assessment of this coarsened graph is often quantified using the node profile matrix.

The node profile matrix of a coarsened graph, denoted as $\phi$, is defined as :

$$\phi = C^T Y \qquad (6)$$

Here, $Y \in \mathbb{R}_+^{p \times l}$ represents the one-hot label matrix of the original graph. In the matrix $\phi$, each non-zero entry $\phi_{ij}$ signifies the count of nodes from the original graph with the $j^{th}$ label that are mapped to the $i^{th}$ supernode in the coarsened graph $\mathcal{G}_c$. A balanced mapping is characterized by the sparsity of each row in the $\phi$ matrix, indicating that nodes with similar labels from the original graph are effectively mapped into a supernode of the coarsened graph.

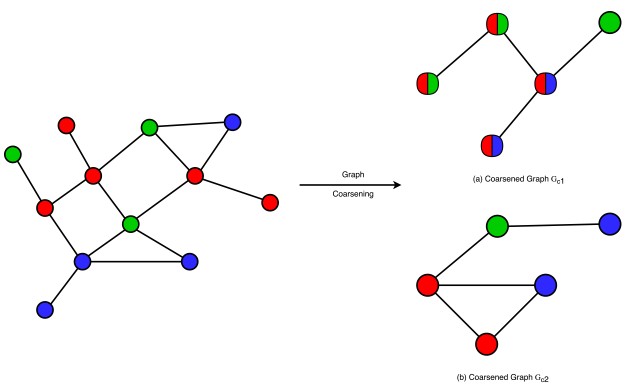

Figure 1: This illustration illustrates that, for a given original graph $\mathcal{G}$, there exist numerous possibilities for the coarsened graph. Notably, it is discernible that the coarsened graph $\mathcal{G}_{c2}$ stands out as a more informative representation as similar label nodes get mapped to the same supernode, making it particularly well-suited for downstream tasks.

Let's consider an toy example in the Figure 1 involving an original graph $\mathcal{G}$ and examine two coarsened graphs, each with its associated $\phi$ matrix:

$$[\phi_1] = \begin{bmatrix} 2 & 1 & 0 \\ 1 & 1 & 0 \\ 0 & 1 & 0 \\ 1 & 0 & 3 \\ 0 & 0 & 1 \end{bmatrix} \qquad \phi_2 = \begin{bmatrix} 3 & 0 & 0 \\ 2 & 0 & 0 \\ 0 & 3 & 0 \\ 0 & 0 & 3 \\ 0 & 0 & 1 \end{bmatrix}$$

The increased sparsity observed in $\phi_2$ compared to $\phi_1$ is indicative of a more pronounced trend: nodes with similar labels are consistently mapped to the same supernode. This, in turn, suggests that the coarsened graph corresponding to $\phi_2$ encapsulates a more focused and distinctive representation of the original graph compared to the coarsened

graph associated with $\phi_1$. Consequently, when contemplating downstream tasks, leveraging the coarsened graph derived from $\phi_2$ proves more advantageous, as it is not only more informative but also specifically tailored to capture essential structural and label-related characteristics of the original graph.

Furthermore, current graph coarsening techniques face limitations in effectively learning coarsened graphs when the associated $\phi$ matrix is sparse. This constraint hinders their suitability for downstream tasks, particularly node classification using the coarsened graph. Additionally, these existing methods often overlook the label information inherent in the original graph during the coarsening process. Consequently, the resulting coarsened graphs lack crucial information, leading to suboptimal performance in downstream tasks. To address these challenges, there is a need for more advanced graph coarsening techniques that incorporate graph matrix, feature matrix and label information of the original graph while doing the coarsening such that the learned coarsened graph is more informative and having sparse $\phi$ matrix.

In the subsequent section, we introduce the first optimization-based approach that consider graph matrix, feature matrix and label matrix of the original graph as inputs, aiming to learn a more informative coarsened graph characterized by a sparse $\phi$ matrix. Notably, during the coarsening process, we selectively taken the information of label in a semisupervised manner.

## 3.2 PROPOSED FORMULATION

Given a graph $\mathcal{G}(\Theta \in \mathbb{R}^{p \times p}, X \in \mathbb{R}^{p \times n}, Y)$, where $Y \in \{0, 1\}^{p \times l}$, the label matrix $Y$ follows a binary encoding, with $\mathbf{y}_i$ : representing the corresponding one-hot indicator vector if node $v^i$ is labeled; otherwise, $\mathbf{y}_i := 0$ for unlabelled nodes in a semi-supervised fashion. The proposed formulation for learning a coarsened graph, emphasizing sparsity in the $\phi$ matrix, is as follows:

$$\min_{\Theta_c, \tilde{X}, C} -\gamma \log \det(\Theta_c + J) + \mathrm{tr}(\tilde{X}^T \Theta_c \tilde{X}) \qquad (7)$$

$$+ \beta h(\Theta_c) + \frac{\lambda}{2} g(C) + r(C, Y)$$

$$\text{s.t. } C \geq 0, \ \Theta_c = C^T \Theta C, \ X = C\tilde{X}, \ \Theta_c \in \mathcal{S}_\Theta, C \in \mathcal{C}$$

In this work, we have opted for $r(C, Y) = \|C^T Y\|_F^2$ as our guiding function. This particular formulation is designed to enforce sparsity within the $\phi$ matrix of the coarsened graph. Furthermore, the term $-\log \det(\Theta_c + J)$ ensure the connectedness of the coarsened graph where, $J = \frac{1}{k}\mathbf{1}_{k \times k}$ is a rank 1 matrix with each element equals to $\frac{1}{k}$. On the other hand, the original hard constraint $X = C\tilde{X}$ poses challenges in optimization. To address this, we relax $X = C\tilde{X}$ to $\|C\tilde{X} - X\|_F^2$ and introduce regularizers $h(\Theta_c) = \|\Theta_c\|_F^2$ and $g(C) = \|C^T\|_{1,2}^2$. Putting $\Theta_c = C^T \Theta C$ in

equation (7) three-variable optimization problem converted into two variable optimization problem as:

$$\min_{\tilde{X},C} -\gamma \log \det(C^T\Theta C + J) + \text{tr}(\tilde{X}^T C^T\Theta C\tilde{X}) \quad (8)$$

$$+\frac{\alpha}{2}\|C\tilde{X} - X\|_F^2 + \frac{\lambda}{2}\|C^T\|_{1,2}^2 + \frac{\beta}{2}\|C^T\Theta C\|_F^2 +$$

$$\frac{\delta}{2}\|C^T Y\|_F^2$$

$$\text{s.t. } \mathcal{S}_C = \left\{C \geq 0\mid \|[C^T]_i\|_2^2 \leq 1 \,\forall\, i = 1,..,p\right\}$$

where, term $\frac{\beta}{2}\|C^T\Theta C\|_F^2$ is incorporated to enforce sparsity in the learned coarsened graph. Meanwhile, the term $\frac{\delta}{2}\|C^T Y\|_F^2$ plays a crucial role in promoting sparsity within the $\phi$ matrix. This sparsity condition ensures that nodes sharing the same label are consistently mapped to the same supernode, thereby enhancing the coherence and consistency of the mapping process.

The proposed formulation (8) is a non-convex optimization problem when considering all variables simultaneously. However, the problem transforms into a convex optimization problem when isolating one variable at a time, treating the remaining variables as constants. Our objective is to address this problem iteratively using a block successive upper bound minimization (BSUM) approach and develop a block MM-based algorithm. This algorithm updates one variable at a time while keeping the other constant, leading to a more manageable and convergent optimization process for the variables $(\tilde{X}, C)$.

### 3.3 UPDATE OF $C$

When considering $C$ as a variable and holding $\tilde{X}$ constant, the resulting sub-problem for $C$ can be expressed as follows:

$$\min_{C\in\mathcal{S}_c} f(C) = -\gamma \log \det(C^T\Theta C + J) + \frac{\lambda}{2}\|C^T\|_{1,2}^2 \quad (9)$$

$$+\frac{\alpha}{2}\|C\tilde{X} - X\|_F^2 + \text{tr}(\tilde{X}^T C^T\Theta C\tilde{X}) + \frac{\beta}{2}\|C^T\Theta C\|_F^2$$

$$+\frac{\delta}{2}\|C^T Y\|_F^2$$

The functions $-\gamma\log\det(C^T\Theta C + J)$, $\frac{\lambda}{2}\|C^T\|_{1,2}^2$, $\frac{\alpha}{2}\|C\tilde{X} - X\|_F^2$, and $\text{tr}(\tilde{X}^T C^T\Theta C\tilde{X})$ are all convex functions [Kumar et al., 2023a]. Additionally, the terms $\frac{\beta}{2}\|C^T\Theta C\|_F^2$ and $\frac{\delta}{2}\|C^T Y\|_F^2$ involve Frobenius norms, rendering them convex functions as well. Considering the set $\mathcal{S}_c$ as a closed convex set, it can be asserted that the optimization problem (9) is strictly convex.

By using the first-order Taylor series approximation, a majorised function for $f(C)$ at $C^{(t)}$ can be obtained as [Beck

and Pan, 2018, Razaviyayn et al., 2012, Sun et al., 2017]:

$$g(C|C^{(t)}) = f(C^{(t)}) + (C - C^{(t)})\nabla f(C^{(t)}) + \frac{L}{2}\|C - C^{(t)}\|^2 \quad (10)$$

where $f(C)$ is $L$−Lipschitz continuous gradient function $L = \max(L_1, L_2, L_3, L_4, L_5, L_6)$ with $L_1, L_2, L_3, L_4, L_5, L_6$ the Lipschitz constants of $-\gamma\log\det(C^T\Theta C + J)$, $\text{tr}(\tilde{X}^T C^T\Theta C\tilde{X})$, $\|C\tilde{X} - X\|_F^2$, $\|C^T\|_{1,2}^2$, $\frac{\beta}{2}\|C^T\Theta C\|_F^2$, $\frac{\delta}{2}\|C^T Y\|_F^2$ respectively. After ignoring the constant term, the majorised problem of (9) is

$$\underset{C\in\mathcal{S}_c}{\text{minimize}} \quad \frac{1}{2}C^T C - C^T A \quad (11)$$

where $A = C^{(t)} - \frac{1}{L}\nabla f(C^{(t)})$ and $\nabla f(C^{(t)}) = -2\gamma\Theta C^{(t)}(C^{(t)^T}\Theta C^{(t)} + J)^{-1} + \alpha\left(C^{(t)}\tilde{X} - X\right)\tilde{X}^T + 2\Theta C^{(t)}\tilde{X}\tilde{X}^T + \lambda C^{(t)}\mathbf{1} + \beta\Theta CC^T\Theta C + \delta Y(C^\top Y)^\top$ where $\mathbf{1}$ is all ones matrix of dimension $k \times k$.

**Lemma 1** *By using KKT optimality condition we can obtain the optimal solution of* (11) *as*

$$C^{(t+1)} = \left(C^{(t)} - \frac{1}{L}\nabla f\left(C^{(t)}\right)\right)^+ \quad (12)$$

*where* $(X_{ij})^+ = \max(\frac{X_{ij}}{\|[X^T]_i\|_2}, 0)$ *and* $[X^T]_i$ *is the i-th row of matrix $X$.*

*Proof:* The proof is deferred to the Appendix A.

### 3.4 UPDATE OF $\tilde{X}$

Fixing $C$, we obtain the following problem for $\tilde{X}$:

$$\min_{\tilde{X}} f(\tilde{X}) = \text{tr}(\tilde{X}^T C^T\Theta C\tilde{X}) + \frac{\alpha}{2}\|C\tilde{X} - X\|_F^2 \quad (13)$$

The problem (13) is a strongly convex optimization problem as $C^T\Theta C$ and $C^T C$ are the positive semi-definite and definite matrices, respectively. The closed form solution of problem (13) can be obtained by setting the gradient to zero, i.e., $2C^T\Theta C\tilde{X} + \alpha C^T(C\tilde{X} - X) = 0$, we get

$$\tilde{X}^{t+1} = \left(\frac{2}{\alpha}C^T\Theta C + C^T C\right)^{-1} C^T X \quad (14)$$

It is noteworthy that the label matrix $Y$ is structured as a one-hot matrix, representing the labels of the nodes in the original graph $\mathcal{G}$ in a semi-supervised fashion. Each row of $Y$ corresponds to a node, and the one-hot encoding signifies the presence of a specific label for that node.

**Theorem 1** *The sequence* $\{C^{(t)}, \tilde{X}^{(t)}\}$ *generated by Algorithm 1 converges to the set of Karush–Kuhn–Tucker (KKT) points of Problem* (8).

*Proof:* The proof is deferred to the Appendix B.

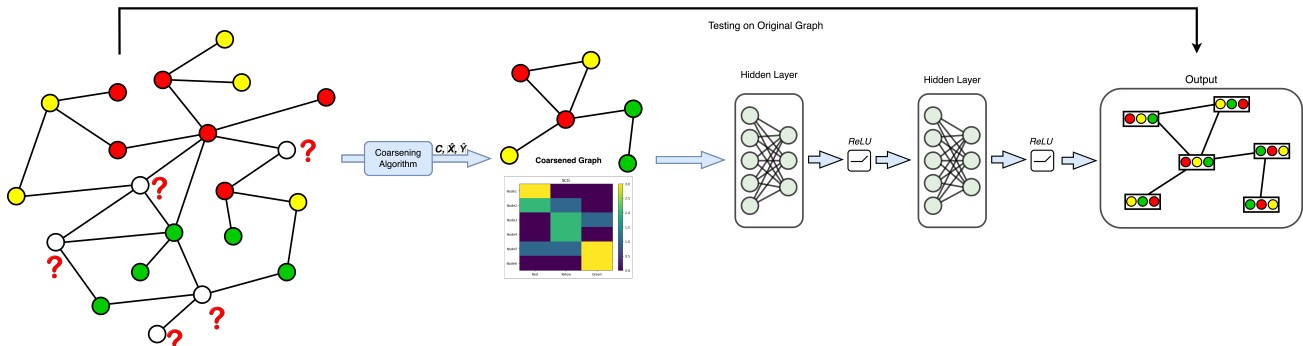

Figure 2: The diagram above illustrates the sequence of steps in performing node classification task using a coarsened graph. Given an original graph $\mathcal{G}(\Theta, X, Y)$ where some of the node labels are known, we employ the LAGC algorithm learn a coarsened graph characterized by a sparser $\phi$ matrix. The resulting coarsened graph is denoted as $\mathcal{G}_c(\Theta_c, \tilde{X}, \tilde{Y})$ where, $\tilde{Y} = \mathrm{argmax}(C^\dagger Y)$. Subsequently, the coarsened graph $\mathcal{G}_c$ is utilised to train a Graph Neural Network (GNN). Subsequently, the trained GNN is evaluated by predicting the labels of nodes in the original graph for which labels were not initially known.

---

**Algorithm 1:** LAGC Algorithm

---

**Input:** $\mathcal{G}(X, Y, \Theta), \alpha, \gamma, \lambda, \beta, \delta$
$t \leftarrow 0$;
**while** *stopping criteria not met* **do**
  | Update $C^{t+1}$ and $\tilde{X}^{t+1}$ as in (12) and (14)
  |   respectively.
  | $t \leftarrow t + 1$;
**end**
**Output:** $C, \Theta_c$, and $\tilde{X}$

| Dataset | Nodes | Edges | Features | Classes |
|---|---|---|---|---|
| CORA | 2,708 | 5,429 | 1,433 | 7 |
| CITESEER | 3,327 | 9,104 | 3,703 | 6 |
| DBLP | 17,716 | 52,867 | 1,639 | 4 |
| CO-CS | 18,333 | 163,788 | 6,805 | 15 |
| PUBMED | 19,717 | 44,338 | 500 | 3 |
| CO-PHYSICS | 34,493 | 247,962 | 8,415 | 5 |

Table 1: Overview of the datasets employed for node classification

## 4 EXPERIMENTS

This section presents experiments validating our proposed LAGC algorithm, beginning with experimental settings, followed by a comparative analysis against key baselines, and concluding with a concise demonstration of LAGC's advantages.

### 4.1 EXPERIMENTAL SETUP

**Dataset** We have performed the experiments on the datasets as shown in the Table 1.

**Baseline Techniques:** We validate our algorithm through ex-

tensive experiments on real datasets, benchmarking against state-of-the-art methods: GCOND [Jin et al., 2021], SCAL [Huang et al., 2021], and FGC [Kumar et al., 2023b]. These selections are based on their recent advancements and superior performance, establishing them as leading coarsening approaches.

Next, we have evaluated the performance of our algorithm through node classification accuracy and time taken($\tau$) for coarsening and classification. Experiment with real dataset using our model outperforms all other state-of-the-art method in node classification and time complexity.

### 4.2 NODE CLASSIFICATION

For the node classification task using a coarsened graph, we employ the proposed LAGC algorithm to learn a coarsened graph by considering the original graph $\mathcal{G}(\Theta, X, Y)$, utilising $80\%$ of the original graph's node labels in a semi-supervised manner. After obtaining the coarsened graph $\mathcal{G}_c(\Theta_c, \tilde{X})$, we infer coarsened graph labels using $\tilde{Y} = \arg\max(PY)$, where $P$ denotes the pseudo-inverse of the mapping matrix $C$. Subsequently, a Graph Convolutional Network (GCN) is trained on $\mathcal{G}_c(\Theta_c, \tilde{X}, \tilde{Y})$. Testing is then performed on the remaining $20\%$ of nodes, whose labels were not utilized during coarsening. Moreover, we have also compared the node classification task using the coarsened graph with the task using the original graph. While performing the node classification task using the original graph, we maintained an identical split, utilising $80\%$ of the original graph's node labels for training and the remaining $20\%$ for testing. In the process of node classification, we undertake the following steps:

### 4.3 LINK PREDICTION

We further demonstrate the effectiveness of our proposed LAGC algorithm on downstream tasks like link prediction.

| Data set | r=k/p | GCOND | SCAL | FGC | LAGC | Whole Data |
|---|---|---|---|---|---|---|
| CORA | 0.3 | $81.56 \pm 0.62$ | $79.42 \pm 1.71$ | $84.03 \pm 0.08$ | $\mathbf{87.62 \pm 0.01}$ | |
| | 0.1 | $81.37 \pm 0.40$ | $71.38 \pm 3.62$ | $79.96 \pm 0.18$ | $\mathbf{86.10 \pm 0.03}$ | $89.50 \pm 1.20$ |
| | 0.05 | $78.93 \pm 0.44$ | $55.32 \pm 7.03$ | $77.31 \pm 0.65$ | $\mathbf{82.85 \pm 0.02}$ | |
| CITESEER | 0.3 | $72.43 \pm 0.49$ | $68.87 \pm 1.37$ | $72.85 \pm 0.10$ | $\mathbf{78.51 \pm 1.25}$ | |
| | 0.1 | $70.46 \pm 0.49$ | $71.38 \pm 3.62$ | $69.46 \pm 0.22$ | $\mathbf{76.00 \pm 0.50}$ | $78.09 \pm 1.95$ |
| | 0.05 | $64.03 \pm 2.40$ | $55.32 \pm 7.03$ | $69.02 \pm 0.24$ | $\mathbf{75.70 \pm 0.31}$ | |
| CO-PHYSICS | 0.05 | $93.05 \pm 0.26$ | $73.09 \pm 7.41$ | $93.31 \pm 0.11$ | $\mathbf{94.46 \pm 0.58}$ | |
| | 0.03 | $92.81 \pm 0.31$ | $63.65 \pm 9.65$ | $92.00 \pm 1.78$ | $\mathbf{94.28 \pm 0.21}$ | $96.22 \pm 0.74$ |
| | 0.01 | $92.81 \pm 0.31$ | $63.65 \pm 9.65$ | $91.08 \pm 0.78$ | $\mathbf{93.26 \pm 0.89}$ | |
| PubMed | 0.05 | $78.16 \pm 0.30$ | $72.82 \pm 2.62$ | $78.14 \pm 0.29$ | $\mathbf{82.85 \pm 0.32}$ | |
| | 0.03 | $78.04 \pm 0.47$ | $70.24 \pm 2.63$ | $77.60 \pm 0.16$ | $\mathbf{82.10 \pm 0.21}$ | $88.89 \pm 0.57$ |
| | 0.01 | $77.20 \pm 0.02$ | $50.49 \pm 10.5$ | $76.10 \pm 1.91$ | $\mathbf{81.27 \pm 0.91}$ | |
| CO-CS | 0.05 | $86.29 \pm 0.63$ | $34.45 \pm 10.0$ | $89.12 \pm 0.08$ | $\mathbf{91.36 \pm 0.48}$ | |
| | 0.03 | $86.32 \pm 0.45$ | $26.06 \pm 9.29$ | $86.32 \pm 0.43$ | $\mathbf{90.32 \pm 0.97}$ | $93.32 \pm 0.62$ |
| | 0.01 | $84.01 \pm 0.02$ | $14.42 \pm 8.51$ | $85.41 \pm 0.24$ | $\mathbf{88.27 \pm 0.34}$ | |
| DBLP | 0.05 | $79.15 \pm 0.20$ | $76.52 \pm 2.88$ | $80.08 \pm 0.01$ | $\mathbf{81.64 \pm 0.42}$ | |
| | 0.03 | $78.42 \pm 1.26$ | $75.49 \pm 2.84$ | $79.92 \pm 0.48$ | $\mathbf{80.93 \pm 0.12}$ | $85.35 \pm 0.86$ |
| | 0.01 | $74.29 \pm 0.57$ | $72.01 \pm 1.83$ | $77.47 \pm 0.33$ | $\mathbf{79.49 \pm 0.53}$ | |

Table 2: The table summarizes node classification accuracy on real benchmark datasets for the proposed LAGC algorithm in comparison to GCOND [Jin et al., 2021], SCAL [Huang et al., 2021], and FGC [Kumar et al., 2023a]. For small datasets, coarsening ratios of $r = 0.3$, 0.1, and $r = 0.05$ were considered, while for large datasets, ratios of $r = 0.05$, 0.03, and $r = 0.01$ were used. The proposed algorithm consistently outperforms state-of-the-art methods by a significant margin. Remarkably, on the Citeseer dataset, our method attains a higher node classification accuracy using the coarsened graph compared to the accuracy achieved when the original graph is used for training.

---

**Algorithm 2:** Node Classification using proposed LAGC

**Input:** $\mathcal{G}(\Theta, X, Y)$
**Output:** Trained weight matrix $W^*$
Apply LAGC on $\mathcal{G}$ to learn $P$; $P = C^\dagger$;
Compute feature matrix of the coarsened graph:
  $X' = PX$;
Compute labels of the coarsened graph:
  $Y' = \arg\max(PY)$;
Learn $W^*$ matrix to minimize $\ell(GCN_{G_c}(W^*), Y')$;

---

We evaluated link prediction performance on three citation networks: Cora, Citeseer, and PubMed. In link prediction, the task is to predict the existence of a connection between two nodes. For our link prediction task, we employed the approach of SEAL [Zhang and Chen, 2018]. We split the original graph into training and testing sets, ensuring both sets retain the same number of nodes. The training graph comprises $80\%$ of the original edges, while the remaining $20\%$ are used for testing. We utilize the training graph to learn the coarsened graph and train the graph neural network on this representation. The trained model is then evaluated on the testing graph, with performance measured using the area under the ROC Curve (AUC). Additionally, we compared our model's performance on the link prediction

task with the state-of-the-art FGC algorithm [Kumar et al., 2023b] and the baseline of using the entire graph for prediction. It's important to note that GCOND[Jin et al., 2021] is a deep learning framework designed specifically for node classification tasks and is not suitable for link prediction.

---

**Algorithm 3:** Link Prediction using proposed LAGC

**Input:** $\mathcal{G}(\Theta, X, Y)$
**Output:** GNN model $\mathcal{G}_\theta$
Randomly initialize model parameter $W^*$;
Treat the existing edges as Positive Examples $\mathbf{P}$;
Randomly sample a set of edges to serve as negative examples $\mathbf{N}$;
Divide $\mathbf{P}$ and $\mathbf{N}$ into training and test sets;
Apply LAGC on train set to learn $P$; $P = C^\dagger$;
Update $W^*$ by minimizing binary cross-entropy loss
  $\ell(GNN_{G_C}(W^*), Y'_{u \sim v})$;

---

### 4.4 GENERALIZABILITY OF PROPOSED LAGC ALGORITHM

To demonstrate the generalizability of learning a coarsened graph from our proposed algorithms, we employed various architectures to train the Graph Neural Network (GNN). Specifically, we utilized GNN architectures such as GCN

| Data set | r=k/p | LAGC | FGC | Whole Data |
|---|---|---|---|---|
| Cora | 0.3 | 0.78 | 0.77 | |
| | 0.1 | 0.77 | 0.75 | 0.84 |
| | 0.05 | 0.75 | 0.72 | |
| CITESEER | 0.3 | 0.75 | 0.73 | |
| | 0.1 | 0.74 | 0. 70 | 0.78 |
| | 0.05 | 0.72 | 0.68 | |
| PubMed | 0.05 | 0.77 | 0.67 | |
| | 0.03 | 0.72 | 0.70 | 0.83 |
| | 0.01 | 0.68 | 0.66 | |

Table 3: This table presents the Area Under the ROC Curve (AUC) metric for link prediction using the proposed LAGC algorithm and the state-of-the-art FGC algorithm Kumar et al. [2023a]. The performance is evaluated at various coarsening ratios: $r = 0.3, 0.1$, and $0.05$ for small datasets, and $r = 0.05, 0.03$, and $0.01$ for large datasets. A baseline comparison using the entire dataset is also included. It is evident that the proposed LAGC algorithm outperforms the existing state of the art graph coarsening technique across all coarsening ratios.

[Kipf and Welling, 2016], APPNP [Gasteiger et al., 2018], and GAT[Veličković et al., 2017] for training and executing the node classification task. The table4 illustrates that our methods for learning the coarsened graph are compatible with different widely used GNN architectures, yielding nearly identical node classification accuracy obtained on different GNN structures.

| Data set | GCN | GAT | APPNP |
|---|---|---|---|
| Cora | $84.45 \pm 0.1$ | $80.23 \pm 0.2$ | $86.05 \pm 0.4$ |
| Citeseer | $75.61 \pm 0.6$ | $72.72 \pm 0.9$ | $76.40 \pm 0.2$ |
| Pubmed | $80.91 \pm 0.1$ | $73.92 \pm 0.2$ | $79.62 \pm 0.6$ |
| Co-CS | $88.27 \pm 0.3$ | $84.49 \pm 0.0$ | $90.27 \pm 0.2$ |

Table 4: Node classification accuracy (%) obtained using different GNN structures like GCN [Kipf and Welling, 2016], GAT [Veličković et al., 2017], and APPNP [Gasteiger et al., 2018]. The experiments were conducted on various datasets, employing the LAGC algorithm with a coarsening ratio of 0.1 for Cora and Citeseer datasets and 0.01 for PubMed and Coauthor CS datasets. It is evident that the proposed LAGC method is suitable for all GNN architecture.

## 4.5 NODE PROFILE MATRIX

To quantify the coarsened graph quality betweenthe proposed LAGC algorithm and state-of-the-art methods, we computed the mapping matrix $C$ and derived the corresponding node profile matrix $\phi = C^\top Y$. Upon comparing the heat maps of $\phi$ matrices with the recent FGC algorithmKumar et al. [2023a], we observed that the LAGC-generated $\phi$ ma-

trix is significantly sparser. This sparsity indicates a higher-quality coarsened graph produced by LAGC compared to FGC [Kumar et al., 2023a]. Also, note that our comparison was made with FGC solely because GCOND [Jin et al., 2021] cannot learn the mapping matrix during the coarsening process.

**Misclassified labels:** The coarsened graph labels, denoted as $\tilde{Y}$, are determined by selecting the class index that maximises the corresponding entry in the product $C^\dagger Y$ where $Y$ is some node label matrix of the original graph. The misclassified label for each supernode $i$ (where $i = 1, 2, \ldots, k$) is computed as the sum of all non-zero entries in the $i^{th}$ row of $\phi$ matrix, excluding the maximum entry. The total misclassified labels, represented as $q$, are the summation across all nodes.

In our comparative analysis, we evaluate the performance by quantifying the number of misclassified labels ($q$). We computed misclassified labels for the Cora dataset with coarsening ratios of 0.05 and 0.1. The state-of-the-art FGC [Kumar et al., 2023a] algorithm resulted in 250 and 458 misclassified points, while our proposed LAGC algorithm yielded 180 and 338 misclassified points for the respective coarsening ratios. The LAGC algorithm exhibits superior performance with fewer misclassifications.

Moreover, the heat map in the Figure 5, depicting $\phi$ matrices from both the proposed LAGC and state-of-the-art FGC [Kumar et al., 2023a] algorithm, provides a visual confirmation of the efficacy of LAGC. It vividly illustrates a notable reduction in misclassified points compared to the FGC algorithm.

## 4.6 RUN-TIME COMPLEXITY:

Given an input graph with $p$ nodes, $E_1$ edges, and a feature vector of size $n$ for each node, the time complexity for node classification using a Graph Convolutional Network (GCN) with $l$ layers is $\mathcal{O}(lp^2n + lpE_1n)$ [Blakely et al., 2021].

The worst-case per iteration computational complexity of our proposed LAGC algorithm is $\mathcal{O}(p^2k)$ for learning a coarsened graph. However, when both coarsening and node classification are performed, the overall time complexity of our algorithm is $\mathcal{O}(p^2k + lk^2n + lkE_2n)$, where $k$ represents the number of nodes in the coarsened graph, and $E_2$ is the number of edges in the coarsened graph. Given that $(p >> k)$ and $E_1 >> E_2$, and choosing $k$ such that $k < n$, the time complexity for coarsening and node classification becomes significantly lower compared to performing node classification solely on the original graph. The effectiveness of this approach is evident in Table 5, demonstrating that the proposed LAGC algorithm is notably faster than baseline methods and exhibits similar time complexity compared to the FGC algorithm [Kumar et al., 2023a,b].

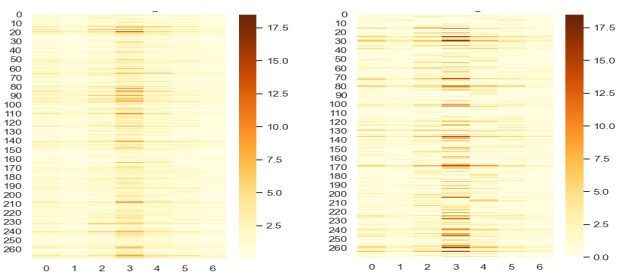

Figure 3: $\phi$ matrix (LAGC)        Figure 4: $\phi$ matrix (FGC)

Figure 5: In Figure (3) to (4), we present heat maps of the $\phi$ matrix obtained from our proposed LAGC and the state-of-the-art FGC algorithm [Kumar et al., 2023a]. Notably, the $\phi$ matrix derived from our algorithm exhibits greater sparsity compared to FGC, highlighting the effectiveness of our approach. Furthermore, the number of misclassified labels ($q$) is 338 and 458 for a coarsening ratio of 0.1 for the proposed LAGC and the state-of-the-art FGC algorithm [Kumar et al., 2023a], respectively. This contrast illustrates that the coarsened graph learned from our proposed algorithm has higher quality than the coarsened graph learned from the state-of-the-art method.

| Dataset($\tau$) | GCOND | SCAL | FGC | LAGC |
|---|---|---|---|---|
| Cora | 329.8 | 27.7 | 1.71 | 1.55 |
| Citeseer | 331.3 | 56.2 | 2.15 | 2.03 |
| Pubmed | 202.0 | 54.0 | 19.81 | 20.35 |
| Co-CS | 1600 | 180 | 34.4 | 49.87 |

Table 5: The table presents a time complexity analysis comparing the proposed LAGC algorithm with baseline algorithms GCOND[Jin et al., 2021], SCAL [Huang et al., 2021], and FGC [Kumar et al., 2023a], considering a coarsening ratio of $r = 0.05$, where $\tau$(in sec.) is the time required to perform coarsening and classification. It is evident that the proposed LAGC is much faster than the existing baselines and comparable to FGC algorithm [Kumar et al., 2023a].

## 5  CONCLUSION

This paper introduces the Label-Aware Graph Coarsening (LAGC) algorithm, an optimization-based approach that uniquely incorporates the graph matrix, feature matrix, and label matrix of the original graph for coarsening, marking the first algorithm to consider the label matrix during this process. The method, efficiently solved through block successive upper bound minimization (BSUM), iteratively updates variables while ensuring convergence. Extensive experimentation demonstrates LAGC's superior performance in node classification tasks compared to state-of-the-art methods, establishing a significant advancement in graph coarsening techniques. LAGC's contributions include a comprehensive optimization strategy, an efficient algorithm, and

empirical evidence highlighting its practical advantages in enhancing graph-based analyses.

## 6  ACKNOWLEDGEMENTS

We would like to thank the editor and reviewers for their suggestions to improve this paper. We would also like to thank to Mr. Prakash Pal for his contribution to the discussions regarding the paper and code implementation. This work is supported by the DST Inspire faculty grant MI02322G.

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

# Supplementary Material

**Manoj Kumar**[*,1]      **Subhanu Halder**[*,1]      **Archit Kane**[2]      **Ruchir Gupta**[2]      **Sandeep Kumar**[1]

[1]Dept. of Electrical Engineering, Indian Institute of Technology Delhi, India
[2]Dept. of Computer Science and Engineering, Indian Institute of Technology (BHU) Varanasi, India

## A   PROOF OF LEMMA 1

The Lagrangian function of (12) is

$$L(C, \tilde{X}, \boldsymbol{\mu}_1) = \frac{1}{2} C^\top C - C^\top A - \boldsymbol{\mu}_1^\top C \tag{15}$$

where $\boldsymbol{\mu}_1$ is the dual variable. The KKT conditions of (12) is

$$C - A - \boldsymbol{\mu}_1 = 0, \tag{16}$$
$$\boldsymbol{\mu}^\top C = 0, \tag{17}$$
$$C \geq 0, \tag{18}$$
$$\boldsymbol{\mu}_1 \geq 0 \tag{19}$$

The optimal solution of $C$ that satisfies all KKT conditions (16-19) is

$$C^{t+1} = (A)^+ \tag{20}$$
$$= \left( C^{(t)} - \frac{1}{L} \nabla f \left( C^{(t)} \right) \right)^+ \tag{21}$$

This concludes the proof.

## B   PROOF OF THEOREM 1

The Lagrangian function of (8) is

$$L(C, \tilde{X}, \boldsymbol{\mu}_1, \boldsymbol{\mu}_2) = -\gamma \log \det(C^T \Theta C + J) + \frac{\alpha}{2} \|X - C\tilde{X}\|_F^2 + \text{tr}(\tilde{X}^T C^T \Theta C \tilde{X}) + \frac{\lambda}{2} \sum_{i=1}^{p} \|[C^T]_i\|_1^2 \tag{22}$$

$$+ \frac{\beta}{2} \|C^T \Theta C\|_F^2 + \frac{\delta}{2} \|C^T Y\|_F^2 - \text{tr}(\boldsymbol{\mu}_1^\top C) + \boldsymbol{\mu}_2^T \left[ (\|C_1^T\|_2^2 - 1), \dots, (\|C_p^T\|_2^2 - 1) \right]^T + \frac{\delta}{2} \|C^T Y\|_F^2$$

where $\boldsymbol{\mu}_{1(p \times k)}$ and $\boldsymbol{\mu}_{2(p \times 1)}$ are the dual variables.
The KKT conditions with respect to $C$ is

(i) $-2\gamma \Theta C(C^T \Theta C + J)^{-1} + \alpha \left( C\tilde{X} - X \right) \tilde{X}^T + 2\Theta C \tilde{X}\tilde{X}^T + \lambda C \mathbf{1}_{k \times k} + \delta Y (C^\top Y)^\top - \boldsymbol{\mu}_1 + 2\left[ \mu_{21} C_1^T, \dots \mu_{2p} C_p^T \right]^T + 2\beta \Theta C C^T \Theta C = 0,$

(ii) $\{\mu_{2i}(\|C_i^T\|_2^2 - 1) = 0\}_{i=1}^p, \quad \text{tr}(\boldsymbol{\mu}_1^\top C) = 0$

(iii) $C \geq 0, \; \|[C^T]_i\|_2^2 \leq 1 \; \forall \, i = 1, 2, \ldots, p,$

(iv) $\boldsymbol{\mu}_1 \geq 0, \quad \boldsymbol{\mu}_2 \geq 0$

where $\mathbf{1}_{k \times k}$ is a $k \times k$ matrix whose all entry is one. The variable $C$ is derived by using the KKT condition from (12):

$$C^\infty - C^\infty + \frac{1}{L}\Big( -2\gamma\Theta C^\infty \big((C^\infty)^T\Theta C^\infty + J\big)^{-1} + \alpha(C^\infty\tilde{X}^\infty - X)(\tilde{X}^\infty)^T + 2\beta\Theta C^\infty(C^\infty)^T\Theta C^\infty \tag{23}$$

$$+2\Theta C^\infty\tilde{X}^\infty(\tilde{X}^\infty)^T + \lambda C^\infty\mathbf{1}_{k \times k} + \delta Y\big((C^\infty)^\top Y\big)^\top\Big) = 0$$

For $\boldsymbol{\mu}_1 = 0$ and $\mu_{2i}[C^T]_i^\infty = 0 \; \forall \, i = 1, 2, \ldots p$, we observe that $C^\infty$ satisfies the KKT condition.
The KKT condition with respect to $\tilde{X}$ is

$$2C^T\Theta C\tilde{X} + \alpha C^T(C\tilde{X} - X) = 0$$

The solution of convex optimization problem (13) is

$$\tilde{X}^\infty = \Big(\frac{2}{\alpha}(C^\infty)^T\Theta C^\infty + C^T C\Big)^{-1}(C^\infty)^T X^\infty$$

which satisfies the KKT condition. This concludes the proof.