# OpenReview forum: "Optimization Framework for Semi-supervised Attributed Graph Coarsening"
_auai.org/UAI/2024/Conference — UAI 2024 poster_

### Official Review · Reviewer_eT41 · 2024-03-20

**Q2-1 Originality-Novelty:** 4
**Q2-2 Correctness-Technical Quality:** 4
**Q2-5 Clarity Of Writing:** 4

**Q1 Summary And Contributions:**

This paper introduces the Label-Aware Graph Coarsening algorithm to enhance the coarsening process by incorporating label information from the original graph. The main contribution of this paper is proposing an optimization-based approach that considers the graph matrix, feature matrix, and label matrix during coarsening, leading to improved performance in downstream tasks like node classification. Experiments show the superior performance of LAGC in node classification tasks compared to state-of-the-art methods.

**Q2-3 Extent To Which Claims Are Supported By Evidence:**

4: Excellent: all claims are supported by very convincing evidence (in the form of comprehensive experimental evaluation, rigorous mathematical proofs, detailed (pseudo-)code, precise references, well-motivated and realistic assumptions) and the authors deliver what they promise.

**Q2-4 Reproducibility:**

3: Good: key resources (e.g. proofs, code, data) are available and key details (e.g. proofs, experimental setup) are sufficiently well-described for competent researchers to confidently reproduce the main results.

**Q3 Main Strengths:**

1.	This paper introduces the Label-Aware Graph Coarsening (LAGC) algorithm. LAGC is the first algorithm to consider the label matrix during the graph coarsening process, contributing to a more informative coarsened graph.
2.	This paper introduces the Block Successive Upper Bound Minimization technique (BSUM). This technique guarantees convergence in updating variables, ensuring computational efficiency throughout the algorithm's execution.
3.	Extensive experiments demonstrate the superiority of LAGC in node classification tasks compared to state-of-the-art methods, validating its practical effectiveness.

**Q4 Main Weakness:**

This paper lacks detailed explanations in some parts, such as the color categories of the output node labels in Figure 2, which are not clearly explained.

**Q5 Detailed Comments To The Authors:**

See Q4

**Q9 Complying With Reviewing Instructions:**

Yes

---

> ### Author Rebuttal · Authors · 2024-04-08
>
> We thank the reviewer for accepting our paper. Also, we again thank the reviewer for taking out the time to read our paper and share your reviews. We will try to address all your concerns.
>
> **Regarding color categories of the output node labels in Figure 2**
>
> Figure 2 depicts the node classification task using coarsened graphs through Graph Neural Networks (GNNs). Given an original graph $G(A, X, Y)$, we first learn a coarsened graph using our proposed algorithm. Then, we train the graph neural network on the coarsened graph $G_c(A_c, X_c, Y_c)$ and perform testing on the original graph. The final layer of the GCN employs a softmax activation function, ensuring that the output for each node is a probability vector. All the entries in this vector lie between 0 and 1 and sum equals to 1, with the predicted class corresponding to the element with the highest probability. We aim to illustrate this probability vector in the output node label visualisation. It's important to note that this figure is for illustrative purposes only and does not represent actual model output. If our paper got accepted, we will add these details in the revised manuscript.
>
> We respectfully ask the reviewers to consider increasing our score if our justification addresses your concerns. We would appreciate further feedback if our changes do not address your concerns.

---

### Official Review · Reviewer_dY6F · 2024-03-22

**Q2-1 Originality-Novelty:** 3
**Q2-2 Correctness-Technical Quality:** 3
**Q2-5 Clarity Of Writing:** 3

**Q1 Summary And Contributions:**

This paper proposed a novel method utilise graph matrix, feature matrix, and label matrix of the original graph for learning a coarsened graph. The task is then converted as an optimisation problem via block successive upper bound minimization methods. The authors shown the effectiveness of their method on multiple benchmark datasets.

**Q2-3 Extent To Which Claims Are Supported By Evidence:**

3: Good: the main claims are supported by convincing evidence (in the form of adequate experimental evaluation, proofs, (pseudo-)code, references, assumptions).

**Q2-4 Reproducibility:**

3: Good: key resources (e.g. proofs, code, data) are available and key details (e.g. proofs, experimental setup) are sufficiently well-described for competent researchers to confidently reproduce the main results.

**Q3 Main Strengths:**

The paper is generally well written and proposed methods works well on a wide range of tasks. This paper is well motivated and the logical flow is consistent, the proposed methods seems reasonable with respect to the motivation.

**Q4 Main Weakness:**

Please see my detailed comments in Q5, I think the presentation could be improved a little bit to help reader with weaker background in graph theory. Some of the equations in section 2 and 3 could be elaborated further.

**Q5 Detailed Comments To The Authors:**

In table 2, why pick these particularly coarsening ratios?

And one more question: when the label Y is used in the coarsening process, it means some aspect of the local cluster information is already embedded in the new graph, or in the matrix C. This will for sure leads to better classification results on the original graph, right?

**Q9 Complying With Reviewing Instructions:**

Yes

---

> ### Author Rebuttal · Authors · 2024-04-08
>
> We thank the reviewer for accepting our paper. Also, we again thank the reviewer for taking out the time to read our paper and share your reviews. We will try to address all your concerns.
>
> **Regarding Coarsening Ratios**
>
> We selected these coarsening ratios and the same experimental settings based on existing state-of-the-art methods such as FGC [Kumar et al., 2023a], GCOND [Jin et al., 2021], and SCAL [Huang et al., 2021] to facilitate direct comparisons. However, since our coarsening framework is optimization-based, it offers the flexibility to reduce the graph size as much as desired. It is important to note that significantly reducing the graph size may also decrease node classification accuracy.
>
> **when the label Y is used in the coarsening process, it means some aspect of the local cluster information is already embedded in the new graph, or in the matrix C. This will for sure leads to better classification results on the original graph**
>
> We agreed with the reviewer on this statement. Also this statement is the main motivation of our work.
>
> **Regarding, some of the equations in section 2 and 3 could be elaborated further.**
>
> We thank the reviewer for the suggestion. We will elaborate more regarding the argmax in eq. (3) of the paper. We will add the following statement:
> Equation 3, $\tilde{Y} = \text{argmax}(C^{\dagger}Y)$, denotes the mapping from the original graph label $Y$ to the coarsened graph label $\tilde{Y}$. In this equation, $Y$ represents the one-hot matrix encoding the node labels from the original graph, and $C$ is the mapping matrix learned for the coarsening process. The $\text{argmax}$ operation identifies the index of the maximum value in each row of $C^{\dagger}Y$, corresponding to the label of the supernode in the coarsened graph. For example, consider the $i$-th row of $C^{\dagger}Y$ being [4 3 1 0 0]. This indicates that the $i$-th supernode contains labels from class 1 for 4 nodes, class 2 for 3 nodes, and class 3 for 1 node. After applying the $\text{argmax}$ operation, the label of the $i$-th supernode is determined, which corresponds to the index of the maximum value in the row.
>
> We will add these details to the revised manuscript if our paper is accepted.
>
>
> We respectfully ask the reviewers to consider increasing our score if our justification addresses your concerns. We would appreciate further feedback if our changes do not address your concerns.

---

### Official Review · Reviewer_1M7x · 2024-03-26

**Q2-1 Originality-Novelty:** 2
**Q2-2 Correctness-Technical Quality:** 2
**Q2-5 Clarity Of Writing:** 2

**Q1 Summary And Contributions:**

This paper proposes to define a graph coarsening problem on semi-supervised attribute graph with an optimization framework. The outcome of the optimized graph is used to improve the problem of node classfication. The proposed algorithm aims at the information in label, feature and node level matrices.

**Q2-3 Extent To Which Claims Are Supported By Evidence:**

2: Fair: the main claims are somewhat supported by evidence (but the experimental evaluation may be weak, or does not match entirely with the claims, important baselines may be missing, proofs contain important ideas but lack rigor, algorithmic details are only discussed superficially, references are imprecise, assumptions are not sufficiently motivated or explicated, etc.).

**Q2-4 Reproducibility:**

2: Fair: key resources (e.g. proofs, code, data) are unavailable but key details (e.g. proof sketches, experimental setup) are sufficiently well-described for an expert to confidently reproduce the main results.

**Q3 Main Strengths:**

This paper proposes to study an interesting problem on graph machine learning, where large scale datasets are used commonly in multiple scenarios. The proposed problem aims to solve the data coarsening problem to improve the performance of machine learing models in node classification tasks. Promising results are demonstrated in experimental sections.

**Q4 Main Weakness:**

Since one of the main claims of this paper is on the benefitial of outcomed graphs for classification tasks, it might be better to deliver more comparisons on this aspect in the experimental and theoretic analysis.

**Q5 Detailed Comments To The Authors:**

The algirthms efficiency is lacked in Table 2.  What is the counterpart in Fig 2 is not clear.

**Q9 Complying With Reviewing Instructions:**

Yes

---

> ### Author Rebuttal · Authors · 2024-04-08
>
> We really thank the reviewer  for the valuable feedback. Your comment of “Promising results are demonstrated in experimental section” is very encouraging!.  We will try to address all your concerns.
>
> **Additional Experiments**
>
> As per the suggestion of the reviewer we have performed experiments on more downstream tasks like link prediction to showcase the efficacy of our proposed  LAGC algorithm.
> We have performed link prediction tasks on three citation networks Cora, Citeseer and Pubmed. The aim of link prediction is to predict whether a link between two nodes should exist or not. We have divided the original graph into two separate graphs—one for training and one for testing—with each graph retaining the same number of nodes as the original. The training graph contains 80% of the original graph's edges, while the remaining 20% of the edges are included in the testing graph. Next, we learn the coarsened graph utlizing the training graph and train the graph neural network using this coarsened graph.We evaluate the trained model on the test graph and  performance based on the Area Under the ROC Curve (AUC). We have also used the existing state of the art algorithm FGC[Kumar et al., 2023a] for link prediction tasks and compared our model with  FGC and link prediction using the whole graph. It is important to note that we have compared only with FGC as GCOND[Jin et al., 2021] is a deep learning based framework designed to perform node classification tasks only. GCOND is not designed to perform the link prediction task.
>
> | Dataset   | r| AUC (Proposed LAGC) | AUC(FGC) | AUC(whole dataset)|
> |-|-|-|-|-|
> || 0.3| 0.78| 0.77||
> | Cora| 0.1   | 0.77 | 0.75| 0.84|
> || 0.05| 0.75| 0.72||
> ||||||
> ||0.3| 0.75 | 0.72  ||
> |Citeseer|  0.1  | 0.74 | 0.71|  0.78 |
> || 0.05  | 0.72| 0.68 ||
> ||||||
> || 0.05  | 0.77 | 0.67|  |
> | PubMed| 0.03| 0.72 |0.70|0.83|
> || 0.01| 0.68| 0.66 ||prediction task.
>
> It is evident from above table that the proposed LAGC outperforms the existing graph coarsening method for edge prediction task and obtain comparable accuracy while performing the classification using the original graph.
>
> **Additional Theoretical Aspects**
> Moreover, regarding the theoretical aspect, we will add the $\epsilon-$ similarity between the original graph $G(\Theta, X)$ and $G_c(\Theta_c, \tilde{X})$ in the revised manuscript.
>
> $\epsilon-$similarity:  The coarsened graph data $G_c(\Theta_c ,\tilde{X})$ is $\epsilon$ similar to the original graph data $G(\Theta, X)$, i.e., there exist an $ 0\leq \epsilon \leq 1$ such that the square root of the smoothness of the coarsened graph lies between the $1-\epsilon$ times to $1+\epsilon$ times of the square root of the smoothness original graph. Where smoothness of original graph is defined as $\text{tr}(X^T\Theta X)$  and coarsened graph smoothness is defined as $\text{tr}(\tilde{X}^T\Theta_c \tilde{X})$
>
> Due to the space limitation in the rebuttal, we will provide the proof of $\epsilon$-similarity in the revised manuscript if paper got accepted.
>
> **Regarding Algorithm efficiency in Table2**
>
> Table 2 of the paper shows the node classification classification accuracy on real benchmark datasets for the proposed LAGC algorithm in comparison to GCOND [Jin et al., 2021], SCAL [Huang et al., 2021], and FGC [Kumar et al., 2023a]. For small datasets,
> coarsening ratios of r = 0.3, 0.1, and r = 0.05 were considered, while for large datasets, ratios of r = 0.05, 0.03, and
> r = 0.01 were used. The proposed algorithm consistently outperforms state-of-the-art methods by a significant margin. Also, the proposed method achieves comparable node classification performance to that using the original graph. It is important to note that our algorithm does not lack in performance, as shown in Table 2, where it outperforms existing graph coarsening methods.
>
>
> **Regarding counterpart in Fig 2 is not clear**
>
> Figure 2 depicts the node classification task using coarsened graphs through Graph Neural Networks (GNNs). Given an original graph $G(A, X, Y)$, we first learn a coarsened graph using our proposed algorithm. Then, we train the graph neural network on the coarsened graph and test it on the original graph. The final layer of the GCN employs a softmax activation function, ensuring that the output for each node is a probability vector. All values in this vector lie between 0 and 1 and sum to 1, with the predicted class corresponding to the element with the highest probability. We aim to illustrate this probability vector in the output node label visualisation. It's important to note that this figure is for illustrative purposes only.
>
> We respectfully ask the reviewers to consider increasing our score if our justification addresses your concerns. We would appreciate further feedback if our changes do not address your concerns.

---

### Official Review · Reviewer_McBb · 2024-03-26

**Q2-1 Originality-Novelty:** 2
**Q2-2 Correctness-Technical Quality:** 3
**Q2-5 Clarity Of Writing:** 2

**Q1 Summary And Contributions:**

This paper proposes a semi-supervised graph coarsening method, namely Label-aware graph coarsening (LAGC) algorithm. The algorithm is based on a successive upper bound minimization. The contribution of the paper includes the proposal of the optimization formulation as well as the optimization algorithm, and experimental evaluation.

**Q2-3 Extent To Which Claims Are Supported By Evidence:**

1: Poor: the authors fail to convincingly backup their main claims (e.g., if the experimental evaluation is flawed, proofs are lacking or invalid, references are missing, assumptions are not realistic, not specified, or not motivated).

**Q2-4 Reproducibility:**

3: Good: key resources (e.g. proofs, code, data) are available and key details (e.g. proofs, experimental setup) are sufficiently well-described for competent researchers to confidently reproduce the main results.

**Q3 Main Strengths:**

The proposed formulation and algorithm are technically sound in general.

**Q4 Main Weakness:**

1.	The motivation of the graph coarsening task is weak. Specifically, it is claimed that such a coarsening step helps downstream tasks such as node classification, but I cannot see the necessity of adding an extra coarsening step to current supervised or semi-supervised node classification tasks.
2.	The clarity of the paper can be improved. E.g., in eq 3, which variable is argmax about? In Fig 1, why the left graph can be coarsened to the right ones, and why would G_c2 be better than G_c1? What’s the difference between graph coarsening and clustering?
3.	The experimental result is weak. Specifically, in order to justify that the proposed method is helpful for downstream tasks such as semi-supervised node classification, it should be compared with recent semi-supervised node classification approaches instead of just other coarsening methods. The running time of semi-supervised node classification approaches with/without LAGC should also be reported. It is also necessary to examine the methods on other downstream tasks.

**Q5 Detailed Comments To The Authors:**

See above

**Q9 Complying With Reviewing Instructions:**

Yes

---

> ### Author Rebuttal · Authors · 2024-04-08
>
> Dear reviewer, we thank you for taking out the time to read our paper and share your reviews. We will try to address all your concerns.
>
> A1: Regarding your inquiry into the motivation behind graph coarsening, allow us to assert its pivotal role as a technique for graph dimensionality reduction. Given a graph $G(\Theta, X, Y)$ with p nodes, coarsening methods aim to learn a graph $G_c(\Theta_c, X_c, Y_c)$ with a significantly smaller size, denoted as k, where k << p, while minimizing information loss. This translates to a substantial reduction in complexity—operations previously conducted on larger graphs with O(p^m) complexity (in terms of space, memory, and computational requirements) now require only O(k^m) complexity. This advancement is crucial for the field of graph machine learning, especially considering the proliferation of applications dealing with graphs boasting an unprecedented number of nodes.
> A plethora of works published in top-tier venues underscore the imperative to explore the potential of graph coarsening. The benefits of this approach are multifaceted, including:
>
> **Accessibility**: Coarsening streamlines the analysis and processing of graphs compared to their original counterparts.
>
> **Memory Efficiency**: Coarsened graphs demand significantly less memory for storage. For example, a coarsened graph comprising 1 million nodes may only require approximately 10GB of memory.
>
> **Computational Efficiency**: Leveraging coarsened graphs for downstream tasks reduces the computational need and memory resources necessary for model training.
>
>
> A2:  Equation 3, $\tilde{Y} = \text{argmax}(C^{\dagger}Y)$, denotes the mapping from the original graph label $Y$ to the coarsened graph label $\tilde{Y}$. The $\text{argmax}$ operation identifies the index of the maximum value in each row of $C^{\dagger}Y$, corresponding to the label of the supernode in the coarsened graph.
>
> Figure 1 presents an example of graph coarsening, reducing an original graph with 12 nodes to a coarsened graph with 5 nodes. Generally, graph coarsening techniques can produce a coarsened graph of this size. For downstream tasks like node classification, optimal training of a graph neural network using a coarsened graph is achieved when nodes with similar labels are mapped to the same supernode. This mapping leads to a higher-quality coarsened graph, which in turn implies that the $\phi$ matrix is sparse. Figure 1 depicts two coarsened graphs derived from the same original graph. However, the state-of-the-art methods do not consider the label matrix during the coarsening process, resulting in a $\phi$ matrix that is not sparse as seen in $G_{c1}$.  Our goal is to develop a loss function that ensures the learned coarsened graph has a sparse $\phi$ matrix. Specifically, our formulation uses the $\phi$ matrix as a regularizer to learn a coarsened graph, similar to $G_{c2}$, which exhibits a sparse $\phi$ matrix. $G_{c2}$ is considered better than $G_{c1}$ primarily due to its higher sparsity in the node profile matrix $\phi$.
>
>  Given a set of data points, the clustering aim to segregate groups with similar traits and assign them into clusters.  For community detection, the data points are nodes of a given network. But these methods do not answer how these groups are related to each other. While, coarsening segregates groups with similar traits and assigns them into supernodes, in addition, it also establishes how these supernodes are related to each other. It learns the graph of the supernodes, the edge weights, and finally the effective feature of each supernode. Thus, the scope of the coarsening method is wider than the aforementioned methods.
>
> A3: In this work, we validate the quality of a coarsened graph generated by our proposed algorithm by performing downstream tasks, such as node classification, using the coarsened graph. It is important to note that we are not proposing a new method for semi-supervised learning; rather, we are using graph coarsening as a preprocessing step to reduce the complexity involved in training of the existing graph machine learning technique for the  downstream task node classification techniques. Specifically, models are trained on the coarsened graph instead of the original graph. As per the suggestion of reviewer, we have performed edge prediction task using coarsened graph and it is evident that our proposed method outperforms the existing state of the art method.
>
> | Dataset   | r| AUC (LAGC) | AUC(FGC) | AUC(whole dataset)|
> |-|-|-|-|-|
> || 0.3| 0.78| 0.77||
> | Cora| 0.1   | 0.77 | 0.75| 0.84|
> || 0.05| 0.75| 0.72||
> ||||||
> ||0.3| 0.75 | 0.72  ||
> |Citeseer|  0.1  | 0.74 | 0.71|  0.78 |
> || 0.05  | 0.72| 0.68 ||
> ||||||
> || 0.05  | 0.77 | 0.67|  |
> | PubMed| 0.03| 0.72 |0.70|0.83|
> || 0.01| 0.68| 0.66 ||
>
> We respectfully ask the reviewers to consider increasing our score if our justification addresses your concerns. We would appreciate further feedback if our changes do not address your concerns.

---

### Official Review · Reviewer_Bzc3 · 2024-03-29

**Q2-1 Originality-Novelty:** 2
**Q2-2 Correctness-Technical Quality:** 3
**Q2-5 Clarity Of Writing:** 4

**Q1 Summary And Contributions:**

The work considers the graph coarsening problem in a semi-supervised setting. While the prior work on the problem consider feature context of nodes and graph matrix, the key contribution of the current work is the incorporation of label information into the learning of graph coarsening problem. The demonstrate the efficacy of the approach, experimentally, on a downstream task namely, node classification. They also prove convergence of the algorithm.

**Q2-3 Extent To Which Claims Are Supported By Evidence:**

3: Good: the main claims are supported by convincing evidence (in the form of adequate experimental evaluation, proofs, (pseudo-)code, references, assumptions).

**Q2-4 Reproducibility:**

3: Good: key resources (e.g. proofs, code, data) are available and key details (e.g. proofs, experimental setup) are sufficiently well-described for competent researchers to confidently reproduce the main results.

**Q3 Main Strengths:**

The paper is an excellent from a understandability point of view, especially to someone new to this topic.

It considers an important problem and the idea of leveraging label information, although a natural extension, has been developed well.

**Q4 Main Weakness:**

Novelty is somewhat incremental.

Lack of comprehensive experimental analysis.

**Q5 Detailed Comments To The Authors:**

Novelty is somewhat incremental: the key innovation is the idea of incorporating label information essentially boils down to the inclusion of one additional loss term in the objective, which presents no additional complexity in solving. Even "node profile" based loss term is based on prior works.

My biggest complaint is the lack of comprehensive experimental analysis to demonstrate the efficacy of their approach. The downstream task, which is node classification, seems a bit biased to their approach, since label information is not assumed in the prior works. I think it is important to demonstrate the superiority of the graph coarsening obtained on other downstream tasks that do no involve output label info.

**Q9 Complying With Reviewing Instructions:**

Yes

---

> ### Author Rebuttal · Authors · 2024-04-08
>
> Dear reviewer, we thank you for taking out the time to read our paper and share your valuable insights. We will try to address all your concerns.
>
> **Regarding Novelty**
>
> Our proposed framework presents a significant and to the best of our knowledge the first approach for doing semi-supervised graph coarsening by seamlessly integrating graph matrix and feature matrix along with the available node label information. This integration enhances the coarsening process without introducing additional complexity and without using additional information.  Leveraging the available label information via the node profile matrix, we formulate a tractable loss function, where the optimization of the node profile matrix under sparsity constraints facilitates effective mapping. Notably, our framework marks the first instance of such integration, offering a straightforward solution that incorporates the node profile matrix into the optimization process.
> Through the optimization of the mapping matrix C, our approach effectively groups nodes with similar labels in the coarsened graph, thereby enhancing the quality of the resulting structure. It is important to highlight that our method utilizes the same dataset and same amount of available information as state-of-the-art techniques for node classification.
> For reference we sketch out the work flow of our proposed approach and the existing-state-of-the art approaches. Note that both approaches use same information, just that we are able to obtain a better coarsened graph resulting an improvement in the downstream task.
>
> Given an original graph $G(A,X,Y)$, the goal is to learn the coarsened graph and train the graph neural network using the coarsened graph and finally, perform testing on the original graph.
>
> **Existing state of the art work flow for node classification using coarsened graph**
>
> **Input Graph:**   $G(A, X)$   &rarr; **Learn coarsened graph** $G_c(A_c, X_c, C)$ using $G(A, X)$   &rarr; **Label determination of coarsened graph** :$Y_c = argmax(C^T Y)$ &rarr;**Graph Neural Network training using $G_c(A_c, X_c, Y_c)$:** &rarr; **Testing on Original Graph**
>
> **Our approach for node classification using coarsened graph**
>
> **Input Graph:**   $G(A, X,Y)$   &rarr; **Learn Coarsened graph** $G_c(A_c, X_c, Y_c)$ using $G(A, X,Y) $   &rarr;**Graph Neural Network training using $G_c(A_c, X_c, Y_c)$:** &rarr; **Testing on Original Graph**
>
> **Regarding Node Profile Matrix**
>
> We agree that the node-profile matrix is well utilized and explored indicator matrix for measuring the quality of grouping for example, in clustering [Ghoroghchian et al., 2021]. Yet, surprisingly, this popular matrix, to the best of our knowledge, has not, been harnessed as an optimization variable to refine mapping with desired constraints.  In our proposed work, we not only adapt the node profile matrix for the coarsening task but also integrate it as a regularizer within our optimization-based framework. This innovative approach ensures the sparsity of the node profile matrix, signifying that nodes with similar labels are systematically mapped to the same supernode in the coarsened graph. Notably, while existing state-of-the-art methods leverage node label information during the GNN training process, our approach, termed as LAGC, seamlessly utilizes such information in the coarsened graph learning process, thus propelling the frontier scalability in graph machine learning.
>
> **Additional Experiments**
> As per the suggestion of reviewer, we have performed link prediction task. Link prediction aims to determine whether a link between two nodes should exist. We split the original graph: 80% of edges for training, 20% for testing. We coarsen the training graph, train a neural network on it, and evaluate it using the AUC on the test graph. We compared our results with the FGC algorithm, noting that GCOND only supports node classification, not link prediction.
>
> | Dataset   | r    | AUC using LAGC | AUC using FGC | AUC using whole dataset |
> |-----------|-------|----------------------|--------------------|---------------|
> |           | 0.3   | 0.78                 | 0.77               |               |
> | Cora      | 0.1   | 0.77                 | 0.75               | 0.84        |
> |           | 0.05  | 0.75                 | 0.72               |               |
> |           |       |                      |                    |               |
> |           | 0.3   | 0.75                | 0.72  |        |
> |  Citeseer |  0.1  | 0.74                | 0.71   |  0.78        |
> |           | 0.05  | 0.72               | 0.68 |     |
> |           |  |  | |  |
> |           | 0.05  | 0.77                | 0.67     |  |
> |  PubMed   |  0.03 | 0.72 | 0.70 |  0.83 |
> |           | 0.01  | 0.68| 0.66  | |
>
> We respectfully ask the reviewers to consider increasing our score if our justification addresses your concerns. We would appreciate further feedback if our changes do not address your concerns.

---

### Meta-Review · Area_Chair_U3WQ · 2024-04-19

The paper considers graph-coarsening via an optimization formulation that takes into account label information, when available. This results in compressed graphs that outperform in downstream problems, such as node classification (and link prediction, an evaluation added as part of their rebuttal).

Three reviewers recommend borderline reject, and two reviewers an accept. The main criticisms are that the approach is somewhat straightforward and incremental, and that the evaluation is too narrow. Some minor criticisms concern the writing and proper motivation for the approach.

I take the prior line of work that this paper adds to as evidence that the approach does have its proponents, and that the paper will be of interest to this community. On the evaluation, while the reviewers did not comment on the additional experiments on link prediction that the authors included in their rebuttal, I would expect them to view an additional set of experiments where the method outperforms the state of the art favorably.

In summary, I feel that the authors have addressed the shortcomings identified by some reviewers sufficiently to push the paper over the threshold.